# Comparison of Source Attribution Methodologies for Human Campylobacteriosis

**DOI:** 10.3390/pathogens12060786

**Published:** 2023-05-31

**Authors:** Maja Lykke Brinch, Tine Hald, Lynda Wainaina, Alessandra Merlotti, Daniel Remondini, Clementine Henri, Patrick Murigu Kamau Njage

**Affiliations:** 1Research Group for Foodborne Pathogens and Epidemiology, National Food Institute, Technical University of Denmark, 2800 Kongens Lyngby, Denmark; malbri@food.dtu.dk (M.L.B.); tiha@food.dtu.dk (T.H.); clehen@food.dtu.dk (C.H.); 2Department of Mathematics, University of Padova, 35121 Padova, Italy; lyndanduta.wainaina@studenti.unipd.it; 3Department of Physics and Astronomy, University of Bologna, 40126 Bologna, Italy; alessandra.merlotti2@unibo.it (A.M.); daniel.remondini@unibo.it (D.R.); 4Research Group for Genomic Epidemiology, National Food Institute, Technical University of Denmark, 2800 Kongens Lyngby, Denmark

**Keywords:** source attribution, *Campylobacter*, campylobacteriosis, network analysis, whole-genome sequencing, coherence source clustering, machine learning, Bayesian modelling

## Abstract

*Campylobacter* spp. are the most common cause of bacterial gastrointestinal infection in humans both in Denmark and worldwide. Studies have found microbial subtyping to be a powerful tool for source attribution, but comparisons of different methodologies are limited. In this study, we compare three source attribution approaches (Machine Learning, Network Analysis, and Bayesian modeling) using three types of whole genome sequences (WGS) data inputs (cgMLST, 5-Mers and 7-Mers). We predicted and compared the sources of human campylobacteriosis cases in Denmark. Using 7mer as an input feature provided the best model performance. The network analysis algorithm had a CSC value of 78.99% and an F1-score value of 67%, while the machine-learning algorithm showed the highest accuracy (98%). The models attributed between 965 and all of the 1224 human cases to a source (network applying 5mer and machine learning applying 7mer, respectively). Chicken from Denmark was the primary source of human campylobacteriosis with an average percentage probability of attribution of 45.8% to 65.4%, representing Bayesian with 7mer and machine learning with cgMLST, respectively. Our results indicate that the different source attribution methodologies based on WGS have great potential for the surveillance and source tracking of Campylobacter. The results of such models may support decision makers to prioritize and target interventions.

## 1. Introduction

*Campylobacter* is considered to be an important zoonotic pathogen as it is one of the leading causes of enteric illness in humans globally. In the EU, *Campylobacter* was responsible for more than 120,000 confirmed cases in 2021, and 3740 cases in Denmark [1,2]. These numbers highlight the need for targeted prevention and control, which requires knowledge of sources and transmission routes. Source attribution models link sporadic human cases of a specific foodborne infection to food sources and animal reservoirs by predicting the origin of each human case. Decision makers may use the obtained knowledge to identify and prioritise appropriate food safety interventions and evaluate the impact of these prevention strategies and control programs [3].

Several methods for source attribution are available, including the microbial subtyping approach, comparative exposure assessment, and outbreak summary data, among others. The microbial subtyping approach involves characterizing isolates of specific pathogens by phenotypic and genotypic subtyping methods [3,4,5]. For *Salmonella*, several subtyping methods have been applied over the years, for example serotyping and phage typing, representing the phenotypic methods, and Multiple Locus Variable Number Tandem Repeat Analysis (MLVA), as an example, of a genotypic method [6,7]. In this approach, the distribution of subtypes in isolates from different food and animal sources are compared to the subtype distribution in isolates from humans, which makes it possible to make inferences on the sources of human infections. The assumption is that the distribution of subtypes in the collection of bacterial isolates for each source used in the attribution model reflects the true distribution of subtypes in each source, which requires a representative collection of isolates from each source as well as from humans [5,8,9]. Source attribution using microbial subtyping has proven to be a valuable tool in providing information on the need for initiating food safety initiatives and monitoring the effectiveness of control programs. There are several applications of the microbial subtyping approach for source attribution, for example, the source attribution of *Salmonella* in the Netherlands and Denmark, in which pork and imported chicken along with table-eggs were found to be the major sources of human salmonellosis in Denmark [5,6].

Multilocus sequence typing (MLST) is a common example of the microbial subtyping approach, which has been used to identify lineages in bacterial populations by indexing the variation present in seven housekeeping genes located in various parts of the chromosome [10]. MLST data have previously been used for the source attribution of human C. jejuni infections in the UK, the Netherlands, New Zealand and Denmark [10,11,12]. However, MLST data have limited discriminatory power; therefore, many studies have suggested an alternative of exploiting whole-genome sequencing (WGS), including core genome MLST, k-mer and single-nucleotide polymorphism (SNP) [13,14]. WGS is becoming the preferred approach for the characterization of isolates of foodborne bacteria, as the cost of high throughput sequencing is decreasing and the technique is fairly rapid, reproducible and has a high discrimination power due to its deep resolution. The different bioinformatic outputs (cgMLST, k-mer, and SNP) have proven valuable for exploring the massive and complex WGS data [15].

To further explore and interpret results obtained by the WGS data, different methodologies such as machine learning, or network analysis can be used. The concept of machine learning, where algorithms are trained to search and recognise patterns in complex datasets, is becoming progressively widespread in epidemiological studies and specifically in predictive modelling [16]. Supervised classification models are trained on allelic variations between sources and identify the potential source of human *Campylobacter* infections. The model outcome is the probability for a human case to originate from each of the potential sources included in the model. Machine learning has previously been used for the source attribution of *Salmonella enterica*, *Escherichia coli* and *C. jejuni* [13,15,17,18] and to predict the severity or outcome of microbial infections [19,20,21,22,23].

Recently, network analysis has been demonstrated as an alternative approach for the source attribution of human salmonellosis [14] and campylobacteriosis [9]. The network analysis approach is based on the weighted networks theory, with pairwise distance matrices from source attribution being visualised as fully connected networks. Nodes correspond to *Campylobacter* isolates and links correspond to genetic distances. Weaker links imply a greater genetic distance between isolates. Network communities correspond to groups of vertices with a higher probability of being connected to each other than other members of that group and can be extracted from the fully connected network [24]. The probability of a human isolate being associated with an animal source is computed as the function of the number of links that the human isolate has with other animal isolates. A specific animal source to which human genomes of *Campylobacter* are attributed can also be extracted from the network analysis. Using the network approach, we can identify which structural features of a data set play a fundamental role in determining the internal coherence of clusters [9,14], such as animal sources, species type and year of origin, etc.

Lastly, the Bayesian classification model assigns human cases to sources based on the distribution of subtypes, e.g., clusters from the network analysis, among human cases and the different sources. The method is based on the Hald model, where the number of human cases caused by different subtypes of a pathogen is compared with the distribution of the same subtypes in different animal source foods obtained from the same geographical location [6]. The Bayesian model has been used for the source attribution of several foodborne pathogens, including *Salmonella* spp., ESBL-producing *Escherichia coli* and *Campylobacter* [12,25,26].

In this study, we aim to compare different methodologies for the source attribution of human campylobacteriosis using whole-genome sequencing data from Denmark. We compare the source attribution accuracy of different WGS data inputs—namely, cgMLST, 5-Mers and 7-Mers using Machine Learning, Network Analysis and the Bayesian Approach.

## 2. Materials and Methods

### 2.1. Data Set

We used Danish *Campylobacter* surveillance data and data from projects that were collected between May 2015 and March 2017 and from January 2019 to March 2021. The dataset contained isolates from chicken, cattle, pigs, dogs, ducks, turkeys, humans, bathing water and vegetables. Spices were determined by WGS, and, where species identification was possible, isolates were found to be *Campylobacter jejuni*. The test material was intestinal content (swabs, stools, or appendices), water and various food products. Meat products were collected either in a slaughterhouse or in the retail trade originating from Danish or foreign production. A few isolates from the production environment were also included. Metadata for the animal and food isolates came from the Danish Veterinary and Food Administration (DVFA). The sequenced genomes and metadata were extracted in September 2021. The following information from the databases were received: sample ID, year of collection, sample material, country of origin of the sample, source (host of the *Campylobacter* isolate), and production system. The latter refers to whether the chicken flocks were raised in a conventional, free-range, or organic production system.

Data and metadata on *Campylobacter* isolates from humans were received from Statens Serum Institute’s surveillance from January 2019 to December 2020. Metadata consisted of: sample ID, year of collection, and travel history (yes/no). Isolates from humans with known travel histories were not included in the dataset. The entire data set was screened to remove duplicates and isolates with incomplete metadata.

As the sampled data were collected over a period of seven years, except for 2018, a preliminary step was performed to ensure that there were no major differences over the years among the genetic diversity of the isolates. Therefore, NMDS (Non-Metric Multidimensional Scaling), using MLST results from the entire dataset, was performed to verify that the overall coordinates of isolates from each year were placed similarly. R version 4.0.0 and packages “vegan” and “ecodist” were used to perform NMDS [27,28,29].

### 2.2. Bioinformatics Analysis

All bioinformatic analyses were performed using the Danish National Supercomputer for Life Sciences, Computerome 2.0, a local server for a Linux-based command-line system [30].

#### 2.2.1. cgMLST

Assemblies and cgMLST are described in earlier work [9]. Assemblies were scaffolded and due to the large genetic variability of *Campylobacter*, many allele numbers were missing in the cgMLST data. Missing allele numbers were imputed using the R packages “doParallel” and “missForest” [31,32]

#### 2.2.2. K-mer

The frequency of a set of k-mers is used for identification to gather genomic sequencing data of the same species [33]. K-mer is often used to identify species, as sequences with a high similarity share many k-mers. Normally, k-mers of an unknown source are compared to a reference genome through different databases. In this case, the frequency of each k-mer found in every isolate is used to compare the isolates with each other. The principle of specific k-mer features to recognise species is, in this study, to provide a hypothesis to distinguish isolates from different food and animal sources. Here, it is assumed that *Campylobacter* from the same source share a similar frequency of k-mer. K was established at 5 and 7. Due to the high computational cost, it was not possible to investigate a higher K. K-mers were extracted from assembled genomes using KMC v3.0. [34]. We combined the results for each k in a matrix and used it as input data for the three modelling approaches. The k-mers matrices sum the count of each k-mer present in each of the isolates, with all possible k-mer as column headers and all the isolates as row headers. The matrix was produced using an in-house Python script.

### 2.3. Machine Learning

We applied machine learning algorithms trained on different features to predict the source of human campylobacteriosis cases. Our models are based on the work of [13]. For cgMLST, the models were trained on allelic variations, where features were specific loci in the genome. With k-mer, the models were trained on the frequencies of mers in the genome. We used supervised classification with features and sources (classes) as input and output data, respectively. The modelling was performed with R version 3.6.3 (9 February 2020).

Two preliminary steps were completed: feature reductions and upsampling. We utilised the NearZeroVariance and Boruta function to reduce the number of features, thereby decreasing the complexity of the data and the computation time of the model. The data set showed an unbalanced number of isolates among the various sources. Upsampling was applied to balance the data set and evaluate the influence of an unbalanced data set on the models [35]. Model selection and model construction used the upsampled data, whereas the original data were used for the final predictive model. This study evaluated two supervised classification algorithms, logit boost (LB) and random forest (RF). These algorithms have been applied successfully in studies analysing sequencing data from *Salmonella typhimurium*, Shiga toxigenic *Escherichia coli* and *Listeria monocytogenes* [13,20,21]. The machine learning model consisted of the following three general steps: model selection, model construction and the final predictive model.

The model selection step used only the source data, i.e., data from food and animals, to select and construct the model. The food and animal data set was spilt into two groups, specifically the training and testing data. We randomly generated the training data set, corresponding to 70% of the isolates, and used it to train the models. The remaining 30% of the isolates were used as testing data to assess the performance of the models. We randomly divided the training data set, for ten iterations, into seven subsets to perform cross-validation. The model was built using six subsets of equal size and one held-out subset for prediction. The ability of the model to predict the correct source of the held-out-subset was quantified by the average accuracy. After reporting the average accuracies, the held-out-subset was returned to the training data set. This procedure was repeated until all subsets had been held out and predicted.

For the model construction step, we constructed the models the same way as described for model selection. To evaluate the performance of the model, the following performance indicators were used: valid accuracy, kappa value, confusion matrix, and sensitivity and specificity for each of the sources.

We used the latter performance parameters to select the best algorithm to use as the final model for predicting the source of the human *Campylobacter* cases. As mentioned above, the final model was built on the original not-upsampled food and animal data sets in the same way as described for the model selection and model construction steps.

### 2.4. Network Analysis

The weighted network approach was used in this study, where the pairwise distance matrix was represented as a network with nodes corresponding to human *Campylobacter* isolates and links as a function of the pairwise distance. This pairwise distance was calculated as the number of different cgMLST alleles or k-mer frequency differences between two sequences. The assumption is that genomes from the same source show smaller distances. A fully connected weighted network, with the weight calculated as 1/distance assigned to each link, was built using MATLAB. A threshold was applied such that the resulting binary network nodes were connected by an edge if the weight was greater than the threshold. The threshold was applied to remove weaker links representing larger genetic distances, and it was chosen to maximize the internal coherence of clusters and minimize the number of isolated nodes. Clusters were identified using the thresholding procedure [14].

The best threshold values were obtained using a 70/30 cross-validation procedure on the animal source data and were chosen in order to maximize the internal coherence of clusters (CSC, Equation (Equation 1) [14]) and minimize the number of isolated nodes. The 70/30 cross-validation procedure involved randomly selecting a network training set consisting of 70% of animal origin samples and using this set to obtain a best threshold value. This threshold value was applied to the network constructed using the test set composed of the remaining 30% of the animal samples for the calculation of the CSC, as shown in Equation (Equation 1). This procedure was repeated 100 times and the most frequent threshold value was selected as the best overall threshold for further use in source clustering. The best threshold was then applied to the full pairwise distance matrix consisting of both animal and human isolates such that the human sources could be attributed to specific animal sources [14]. The best threshold was used to maximize the score function on distance matrices, as shown in Equation (Equation 2) [14]. The graphical visualizations of the network were obtained using the MATLAB ‘Plot’ function with the force-directed graph layout [36].
(1)$$CSC=\frac{\sum_{i=1}^{N_{c}}TP_{i}}{\sum_{i=1}^{N_{c}}T_{i}}100$$
(2)$$Score=\left(1-\frac{N_{ISO}}{N_{TOT}}\right)CSC$$

$N_{TOT}$ is the total number of nodes in the network, while $N_{ISO}$ is the number of isolated nodes that do not have any links to other nodes. *CSC* is the coherent source clustering, which measures the algorithm’s clustering performance, where $TP_{i}$ is the number of true positives in the *i*th cluster (majority of isolates from the same source in the same cluster) and $T_{i}$ is the total number of nodes inside the *i*th cluster [14].

### 2.5. Bayesian Classification Model

The clusters from the network analysis were used as the input in a Bayesian classification model described by [6,25]. The expected number of human cases per source and cluster was estimated with the following equation:(3)$$\lambda_{ij}=M_{j}p_{ij}q_{i}a_{j},$$
where $\lambda_{ij}$ is the expected number of cases of cluster *i* caused by source *j*; $M_{j}$ is the total number of source *j* available in the clusters; $p_{ij}$ the prevalence of cluster *i* in source *j*; $q_{i}$ is an unknown cluster-dependent factor; and $a_{j}$ is an unknown source-dependent factor. Briefly, the q-value can be described as an unknown property that should capture any differences between isolates from different clusters with regard to, e.g., infectivity and/or survival during food processing. Similarly, the a-value attempts to capture differences between sources, e.g., variations in monitoring systems and food preparation methods. The $q_{i}$ and $a_{j}$ represent the Bayesian model’s multi-parameter prior distribution. We used a lognormal distribution $log\left(q_{i}\right)∼N\left(0,\tau \right)$, where τ is a parameter controlling the variation in characteristics between types. For τ, the prior distribution was a fairly diffuse gamma prior τ∼gamma
(0.01,0.01). For $a_{j}$, we used an exponential distribution $a_{j}∼exponential\left(\lambda \right)$. The prior distribution for cattle and chicken of unknown origin were set to be equal to cattle and chicken from Denmark, respectively [11].

The expected number of sporadic and domestic cases attributed to each cluster $\lambda_{i}$ was then estimated assuming a Poisson distribution of the observed number of sporadic and domestic cases per cluster $o_{i}$:(4)$$\;o_{i}=Poisson\left(\Sigma_{j}\lambda_{ij}\right)$$ The model was built using the Markov Chain Monte Carlo (MCMC) simulation in R (version R 3.6.3 2020, R Development core team) using the R-packages rjags and coda. Three independent Markov chains with widely dispersed initial values were run for 10,000 iterations after a 3000-iteration burn-in period and with a thinning of 10. Chain convergence was monitored visually by using the methods described by Gelman and Rubin [37] and was considered to have occurred when the variance between the different chains was no larger than the variance within each individual chain, and when the chains had reached a stable level.

## 3. Results and Discussion

We compared three different source attribution methodologies based on whole genome sequencing data. Input data were allele variations derived from the core genome analysis (cgMLST) and frequencies of k-mers (5mer and 7mer). The data were collected between 2015–2021 and consisted of 2590 isolates after data screening and assembly quality assessment. The dataset consisted of 1366 isolates from food, the environment, and animals (2015–2020) and 1224 from human campylobacteriosis cases (2019–2021). Isolates from bathing seawater and vegetables were not included in the modelling, due to the small number of isolates. Additionally, these sources are not considered to be reservoirs for *Campylobacter*, but contaminated from animal sources. We distinguished all non-Danish food or animal sources in the modelling as represented in Table 1. We performed NMDS to assess the variation of broad genetic structures of the sampled isolates over the years (Figure 1). Each isolate is plotted according to its MLST results, and lines are coloured corresponding to the year of sampling. Ellipses encompass all the isolates of the same year and were coloured according to the year. Each ellipse’s centre point is marked the colour of the year it reflects. All centre points of each ellipse were located at similar coordinates: this supports no major changes in ST variation over the different years and it seems valid to pool the samples from these seven years and analyse them together in the modelling.

### 3.1. Performance Comparison of Network Analysis and Machine Learning Models

We compared the accuracy results from machine learning and the network analysis, as shown in Table 2. The accuracy and F1-score values were computed from the confusion matrices. We observed that the accuracy values were quite high and considering the highly imbalanced data sets, F1-scores were used to assess the performance of the models more accurately. The best-performing network analysis algorithm, with 7mer as the input data, had an F1-score of 67%. For the machine learning models, the random forest algorithm showed the highest accuracy with all data inputs, whereas 7mer was the best-performing model with an accuracy of 98%. Arning et al. (2021) demonstrated that the use of cgMLST, compared to the well-established seven-loci MLST, results in more accurate source attribution estimates. When using cgMLST and k-mer, we increase the amount of genomic information the model can use to discriminate between sources, hence the higher accuracy of the models [15]. However, higher accuracy means higher specificity and lower sensitivity, which in general will lead to less attributable human cases [5].

### 3.2. Machine Learning Model

Table 3 shows the confusion matrix for the best performing model—Random Forest with 7mer as input. The model had a kappa value of 0.979, which is a high or almost perfect agreement between observed and predicted cases, as can be seen in the confusion matrix. The sensitivity and specificity were greater than 0.9 in almost all the sources; only the sensitivity for Chicken Denmark was lower (0.831), due to the number of wrongly predicted isolates, e.g., 15 as cattle from Denmark and 17 as foreign chicken. Balanced accuracy is defined as the average accuracy obtained in either class [38]. The balanced accuracy was higher than 0.91 for all sources. The sensitivities for the sources Cattle unknown, Chicken unknown, Dog Denmark, Duck Denmark, Duck foreign, Pig Denmark, Production environment, and Turkey foreign were all one, which appears from the confusion matrix (Table 3), and meaning that for these sources there was 100% agreement between observed and predicted cases. The modelling was based on an imbalanced data set as the number of included isolates varied from 5 to 781 for Cattle unknown and chicken from Denmark, respectively. To overcome this imbalance, the data were upsampled during the model selection and model construction steps. Hence, identical isolates are likely to have appeared in both the training and testing data sets resulting in a general overestimation of the model accuracies for sources with few isolates. Furthermore, human cases will, in general, have a higher probability to be allocated to an intensive sample source because the diversity in the pool of samples from that source is much higher compared to a source where only a few isolates are available. More isolates from these less sampled sources would help us to better evaluate their contribution to the number of human *Campylobacter* infections.

### 3.3. Network Analysis Results

We obtained the best threshold values 0.0228, 0.0019 and 0.0053 for cgMLST, 5mer and 7mer pairwise distance matrices, respectively. These values were used to maximize the score function from 100 runs of cross-validation. The network analysis algorithm attained 78.2%, 78.8% and 79% coherent source clustering for cgMLST, 5mer and 7mer distance matrices, respectively (Table 4).

We computed the confusion matrix for the best-performing model. The model that had a 7mer distance matrix as input data was the best-performing model which was consistent with the results from Table 4. However, some animal isolates were wrongly classified by the algorithm, for example, 21 cattle isolates from Denmark were classified as chicken from Denmark while 99 chicken isolates from Denmark were classified as cattle isolates from Denmark.

The results displayed in Figure 2 show the clustering results from the 7mer distance matrix. Some clusters included two or more isolates from different sources. For example, in clusters 1, 2 and 3 it is observed that chickens from Denmark were classified in the same cluster as dogs from Denmark, cattle from Denmark and chickens from foreign countries. This reflects that these sources may be epidemiologically linked and have a common source of origin, or that the *Campylobacter* strains included in these clusters, besides being very genetically similar, also are more clonally distributed compared to other strains. We observed 100% probability of attribution of some human cases to less abundant sources such as dogs from Denmark, indicating that the algorithm was not heavily influenced by the sample size. The results show that the network analysis algorithm had high specificity due to the number of links between each human isolate and each animal source.

### 3.4. Bayesian Models

A total of 72 clusters (*i*) from the network analysis included isolates from both humans and one or more sources. Ten different source categories (*j*) were found in clusters with human isolates and included in the modelling.

### 3.5. Source Attribution

The models predicted the source origin of 1224 human campylobacteriosis cases in Denmark from 2019 to 2021. Table 5 presents an overview of the source attribution results, with the predicted number of human cases attributed to each source. Chicken from Denmark was predicted as the primary source of human campylobacteriosis independently of the method and input data. The percentage of cases originating from chicken from Denmark varied from 45.8% to 65.4%, for Bayesian with 7mer and machine learning (ML) with cgMLST, respectively. Imported chicken had the second-highest number of cases in the Bayesian models and Machine learning with cgMLST and 7mer. However, cattle from Denmark were predicted as the second highest in the Network analyses and machine learning with 5mer as input data. These results support previous studies which identify chicken as the primary source of human campylobacteriosis in Denmark [9,12,39]. The machine learning models were able to allocate almost all the 1224 isolates from human cases. Network analysis with 5mer could not predict the source of 21.1% of the human isolates. Hereby, the highest number of human isolates was not attributed to any animal source. The Bayesian models did not include 194 of the human cases in the model. These cases were not present in a cluster, where one or more sources were also represented.

A major limitation of using cgMLST as input data is the substantial number of missing alleles, which occurs when no matches can be found in the PubMLST database. This is mainly because of either incomplete assembly or the broad variation in the genome, which result from horizontal gene transfer and mutations. Imputing a substantial amount of missing alleles increases the uncertainty in the results of the final model, but this uncertainty cannot be quantified. Such problems are not encountered when using k-mers as input. However, despite the high accuracy in the 7mer machine learning models, many human cases had a low to medium probability to be allocated to multiple sources. Only 397 (32.4%) of the human cases had a probability above 75% to be allocated to a single source, where the most cases (n = 324) were allocated to chicken from Denmark followed by cattle from Denmark (n = 45). A possible explanation for this might be that the model is not sufficiently discriminatory to distinguish between sources, which again may reflect that some *Campylobacter* strains are widely disseminated and consequently found in several sources and humans. Increasing the number of K will likely result in an increased specificity of the models, as demonstrated by Arning et al. [15]. However, with higher k-mers, the number of variables increases exponentially (k = 8 results in 32,896 variables, k = 9 in 130,646 variables), which makes the running time of the algorithms far from cost-efficient. In addition, higher specificity may actually result in the models being able to allocate fewer human cases to sources, as there will be a lower probability of matching them with isolates from sources. As expected, a number of cases could not be predicted, which to some extent may be explained by this phenomenon, but also because of the inclusion of unreported travel cases or cases of infection by sources not represented in the model, e.g., seafood and wild birds. As mentioned earlier, the Bayesian method used the clusters from the network analysis to create ’subtypes’. This resulted in a number of cases not being included in the modelling, because they were not clustered together with any source isolates. This is in line with what has been described in previous models for, e.g., *Salmonella*, that apply a combination of serovar, phage type and Multiple Locus Variable Number Tandem Repeat Analysis (MLVA) as input data, and may in fact reflect that there always will be human cases that are infected from sources not monitored or by pathogen strains that are not captured during source monitoring [7]. In addition, none of the clusters represented both human cases and isolates from ducks from Denmark, meaning that none of the human cases was attributed to this source.

## 4. Conclusions

The models described in this study can be adapted to other countries, where different sources may be of importance. Further useful work would be to gradually build up a larger training data set by adding isolates over the years, thereby increasing model robustness, especially for machine learning that improves with experience (more data). The results of this study indicate that the different source attribution methodologies based on WGS have great potential for the surveillance and source tracking of *Campylobacter*. The models enable decision makers to prioritise targeted interventions, e.g., in livestock.

## Figures and Tables

**Figure 1 pathogens-12-00786-f001:**
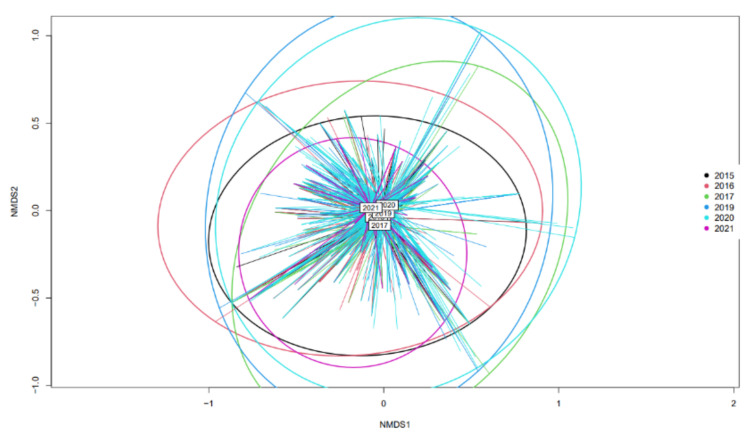
NMDS representation based on MLST dissimilarities using Bray Curtis method. Each isolate is plotted according to its MLST results, and it is represented at the end of each arrow. The colour of each arrow corresponds to the year of sampling of the isolates. Ellipses were built to encompass all the isolates of the same year and were coloured according to the year. The centroid of each ellipse is labelled with the year it represents.

**Figure 2 pathogens-12-00786-f002:**
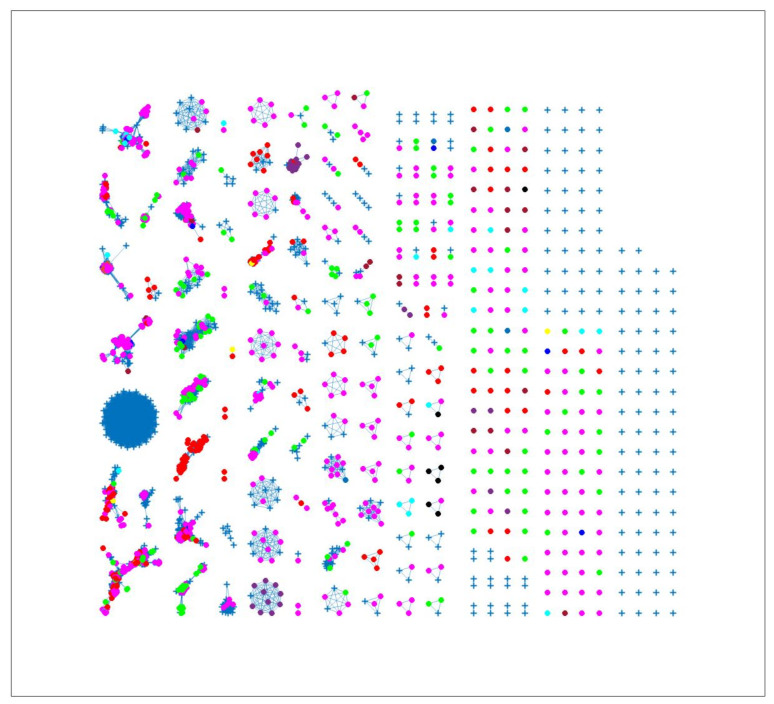
Source clustering results (force-directed graph drawing algorithm) obtained using 7mer distance matrix as model input. Nodes represent different animal isolates. Legend: red—cattle from Denmark; yellow—cattle from unknown countries; magenta—chickens from Denmark; green—chickens from foreign countries; dark blue—chicken from unknown countries; cyan—dogs from Denmark; black—ducks from Denmark; purple—ducks from foreign countries; blue—production environment; light brown—turkeys from foreign countries; dark brown —pigs from Denmark. blue crosses—human isolates. Foreign (Germany, Netherlands, Italy, France, Poland, UK, Hungary).

**Table 1 pathogens-12-00786-t001:** The number of *Campylobacter* samples in the dataset by source and origin. Food, animal, and environment samples were collected in 2015–2017 and 2019–2020. Human samples were collected 2019–2021.

Sources	Origin	Number of Isolates
Human		1224
Chicken	DK	781
Cattle	DK	265
Dog	DK	33
Pig	DK	30
Duck	DK	7
Production Environment	DK	10
Chicken	Imported	184
Duck	Imported	24
Turkey	Imported	9
Chicken	Unknown	18
Cattle	Unknown	5
Total analysed data		2590
Unknown	Unknown	3
Vegetables	DK	3
Seawater	DK	2

**Table 2 pathogens-12-00786-t002:** Accuracy (F1-score) for machine learning models and network analysis with different input data.

	cgMLST	5 Mers	7 Mers
Network analysis	96%(63%)	96%(64%)	96%(67%)
Random Forest	95%	98%(98%)	98%(98%)
Logit Boost	92%	95%(94%)	95%(94%)

**Table 3 pathogens-12-00786-t003:** Confusion matrix for the best machine learning algorithm Random forest and 7mer as input.

Source	Cattle.dk	Cattle.unk	Chkn.dK	Chkn.for	Chckn.unk	Dog.dk	Duck.dk	Duck.for	Pig.dk	Prod	Turkey.for
Cattle.dk	232	0	15	0	0	0	0	0	0	0	0
Cattle.unk	0	249	0	0	0	0	0	0	0	0	0
Chkn.dK	5	0	192	5	0	0	0	0	0	0	0
Chkn.for	0	0	17	218	0	0	0	0	0	0	0
Chckn.unk	0	0	1	0	225	0	0	0	0	0	0
Dog.dk	0	0	3	0	0	255	0	0	0	0	0
Duck.dk	0	0	1	0	0	0	233	0	0	0	0
Duck.for	0	0	0	0	0	0	0	234	0	0	0
Pig.dk	0	0	0	0	0	0	0	0	223	0	0
Prod	0	0	1	0	0	0	0	0	0	235	0
Turkey.for	0	0	1	0	0	0	0	0	0	0	233

Cattle.dk—cattle from Denmark; Cattle.unk—cattle from unknown countries; Chkn.dk—chickens from Denmark; Chkn.for—chickens from foreign countries; Chkn.unk—cattle from unknown countries; Dog.dk—dogs from Denmark; Duck.dk —ducks from Denmark; Duck.for—ducks from foreign countries; Pig.dk—pigs from Denmark; Prod—Production environment; Turkey.for—turkeys from foreign countries.

**Table 4 pathogens-12-00786-t004:** Best threshold and CSC (Coherent source clustering) based on cgMLST, 5mer and 7mer distance matrices.

Performance	cgMLST	5 Mers	7 Mers
Best threshold	0.0228	0.0019	0.0053
CSC	78.18%	78.77%	78.99%

**Table 5 pathogens-12-00786-t005:** The number of human cases predicted for every source (%). Comparison of the source attribution results obtained from the Bayesian model, machine learning model and network analysis.

	Bayesian		Machine Learning			Network Analysis		
	**cgMLST**	**7mer**	**cgMLST**	**5mer**	**7mer**	**cgMLST**	**5mer**	**7mer**
Cattle.dk	99 (8.1)	108 (8.8)	162 (13.2)	217 (17.7)	193 (15.8)	153 (12.5)	153 (12.5)	156 (12.7)
Cattle.unk	2 (0.1)	1 (0.1)	0 (0.0)	6 (0.5)	6 (0.5)	5 (0.4)	1 (0.08)	2 (0.2)
Chkn.dk	601 (49.1)	561 (45.8)	801 (65.4)	668 (54.6)	684 (55.9)	632 (51.6)	583 (47.6)	582 (47.5)
Chkn.for	144 (11.8)	127 (10.4)	174 (14.2)	197 (16.1)	205 (16.7)	103 (8.4)	129 (10.5)	139 (11.4)
Chkn.unk	17 (1.4)	12 (1.0)	0 (0.0)	33 (2.7)	41 (3.3)	34 (2.8)	25 (2.0)	24 (2.0)
Dog.dk	25 (2.0)	29 (2.4)	19 (1.6)	33 (2.7)	31 (2.5)	11 (0.9)	21 (1.7)	18 (1.5)
Duck.dk	0 (0.0)	0 (0.0)	20 (1.6)	2 (0.2)	3 (0.2)	0 (0)	1 (0.1)	0 (0)
Duck.for	15 (1.2)	17 (1.4)	0 (0.0)	20 (1.6)	20 (1.6)	10 (0.8)	10 (0.8)	9 (0.7)
Pig.dk	6 (0.5)	4 (0.3)	7 (0.6)	8 (0.7)	7 (0.6)	3 (0.2)	5 (0.4)	4 (0.3)
Prod	110 (10.7)	105 (8.6)	32 (2.6)	26 (2.1)	23 (1.9)	32 (2.6)	26 (2.1)	28 (2.3)
Turkey.for	10 (0.8)	22 (1.8)	10 (0.8)	12 (1.0)	12 (1.0)	5 (0.4)	11 (0.9)	10 (0.8)
Not estimated	1 (0.1)	44 (3.6)	0 (0.0)	2 (0.2)	0 (0.0)	236 (19.3)	259 (21.2)	252 (20.6)
Not included	194 (15.8)	194 (15.8)	−1 (−0.1) *	0 (0.0)	−1 (−0.1) *	0 (0.0)	0 (0.0)	0 (0.0)

Cattle.dk—cattle from Denmark; Cattle.unk—cattle from unknown countries; Chkn.dk—chickens from Denmark; Chkn.for—chickens from foreign countries; Chkn.unk—cattle from unknown countries; Dog.dk—dogs from Denmark; Duck.dk —ducks from Denmark; Duck.for—ducks from foreign countries; Pig.dk—pigs from Denmark; Prod—Production environment; Turkey.for—turkeys from foreign countries. * Over-prediction due to rounding error.

## Data Availability

Not applicable.

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
