# Peer review of "Comparison of Source Attribution Methodologies for Human Campylobacteriosis"

_pathogens, 2023, doi:10.3390/pathogens12060786_

Round 1

Reviewer 1 Report

Comment to authors

Abstract

The abstract is well written and presented most of the parts, however, the conclusion part should be written based on the results, and the conclusions and their extension to other issues should be written carefully. More details in the abstract are needed for this conclusion: “…can enable decision-makers to prioritize and target interventions”/ or please delete that.

Introduction

Paragraphs 2-3: More details about “the microbial subtyping” methods, used in microbiology and epidemiology studies, should be mentioned. Please give a brief about some previous articles that used a method to find the source of human campylobacters in foods.

Paragraph 4: To further explore and interpret results obtained by the WGS data different methodologies such as Machine learning, or Network Analysis can be used.

Paragraph 4: The method is based on the Hald model, where the number of human cases caused by different subtypes of a pathogen is compared with the distribution of the same subtypes in different animal source foods obtained from the same geographical location.

The introduction is written very well and the importance of the study and its aims are discussed very well.

Materials and Methods

2.1 data set: please add information about the species of isolates in Table 1 along with the number of each isolate. Was there a bacterial species identification method as well as a culture or molecular identification method in the extracted data? Explain this as well. Whether only the identification of the genus was considered in the investigations or the species of bacteria was also examined. What data from human samples was examined?

Result and Discussion

Please give more information of this in the text in the first paragraph of result part: “We performed NMDS to assess the variation of broad genetic structures of the sampled isolates over the years (Figure 1). All centre points of each ellipse were located at similar coordinates….”. More explanations are needed so that the results of the analyzes can be well understood by biologists who are not fluent in computer language or machine learning modeling.

In the following, please provide an offer based on this data: “When using cgMLST and k-mer, we increase the amount of genomic information the model can use to discriminate between sources, hence the higher accuracy of the models [14]. However, higher accuracy means higher specificity and hence lower sensitivity, which in general will lead to less attributable human cases.”

If possible, please mix the data of chicken and cattle (or other sources) regardless of the origin of the animal (Denmark or not) and evaluate the results. It is my suggestion (as all animal sources were in Denmark when the isolates were isolated and could directly play a role in the pathogenesis of humans. Therefore, their cumulative investigation is also useful with all three models, and maybe, for example, in the case of chicken meat, it may even be higher than the reported percentage (65.4 %) as a source of contamination.

References

Good.

Author Response

Dear Reviewer,

Thank you for your feedback!

Reviewer 2 Report

This paper compares three models for disease tracing, the models have been appropriately described. Data is presented well and the subject area is of interest. 

Please check that Campylobacter is in italics throughout your manuscript. 

For the methods and results section  there is first person language use, with the term "we" frequently used, the use of third personal language would be more appropriate.  

Author Response

Dear Reviewer,

Thank you for your feedback. The manuscript is looked through and Campylobacter is corrected to be in italics all times. The method and results section have been reviewed and first-person language is reduced.